# Quality of Life of People with Mobility-Related Disabilities in Sweden: A Comparative Cross-Sectional Study

**DOI:** 10.3390/ijerph192215109

**Published:** 2022-11-16

**Authors:** Karolin Lindgren Westlund, Mats Jong

**Affiliations:** Department of Health Sciences/Public Health, Mid Sweden University, 85170 Sundsvall, Sweden

**Keywords:** cash margin, economic situation, impairment, mobility, social inclusion, WHOQOL-BREF

## Abstract

Little is known about the Quality of Life (QoL) and how QoL is related to the social and economic situation of people with mobility-related disabilities in Sweden. QoL and well-being do not only relate to the absence of impairments but also to the level of social inclusion and the economic situation. The objective of this study was to explore if there were differences in QoL between a group with and a group without mobility-related disabilities in Sweden. Cross-sectional data were collected through self-reported questionnaires. WHOQOL-BREF was used to assess QoL. Recruitment was conducted through social media platforms. Comparisons were made between and within groups using the Welch t-test. Generalized linear models were used to predict score change for the WHOQOL-BREF items and domains accounting for sex, age, education, social inclusion, economic situation, and presence of additional or other disability. Included in the analysis was data from 381 participants, 143 with mobility-related disabilities and 238 without. Participants in the mobility-related disability group scored significantly lower than those without on General Health, General QoL, Health Satisfaction, and the four WHOQOL-BREF domains. The group with mobility-related disabilities also reported a lower Social Inclusion Score (SIS) and a higher proportion of people without a cash margin. An increased SIS indicated higher QoL in the generalized linear model, whereas the absence of cash margin and mobility-related disability negatively influenced the QoL scores. This study indicated that a person with mobility-related disabilities has lower QoL than those without mobility-related disabilities. A lower QoL was also related to a lack of cash margin, a lower social inclusion score, and whether there were additional or other disabilities present.

## 1. Introduction

Disability, in general, is a part of the human experience, and all of us will be temporarily or permanently impaired at some point in our life [1]. A person with a disability is defined as someone who has long-term mental, intellectual, physical, or sensory impairments which, in interaction with different barriers, may hinder their full participation in society on an equal basis with others [2]. There are estimations that fifteen percent of the population globally experience some level of disability. The experience of disability is a complex and multidimensional [1]. Disability is not solely determined by the underlying health condition. The lived experience of disability lies on a continuum from no to complete disability. The experience of disability is influenced by the capability to perform tasks, the facilitators and barriers in the environment, and the health conditions [3]. Impairments or deficits do not necessarily lead to decreased perceived health if limitations in participation and activities can be avoided. The number of persons with impairments who feel healthy and have high performance levels in activity and participation may be increased with proper contextual interventions [4].

People facing impairments and disabilities are a diverse group often found among the least privileged groups [1,5,6]. To belong to a group with fewer privileges influence health determinants such as; assess to, and the quality of health services, leisure time, family and social engagement, participation in the community, work, education, and the ability to influence the overall conditions of life [1,5,6]. Enhancing the health and well-being of all people, including people with disabilities, is crucial to fulfilling Sustainable Development Goals (SDG). Everyone’s health and well-being depend on a satisfactory standard of living, decently paid work, participation in education, and social and community life [7].

Disability and poverty are causes and consequences of each other. There is a bidirectional relationship [5,8]. For poor individuals, the risks of illness, injury, or impairment increase due to living in dangerous and unsanitary conditions, low access to preventive and curative health care services, and unsafe working conditions. People with disabilities and their families have higher risks of poverty and impoverishment because of their low access to education and employment [5].

Social exclusion is associated with low access to fundamental opportunities and social and political participation and directly impacts poverty levels for people with disabilities and their families. Social exclusion is also related to social justice and how minority groups do not have the same opportunities and are discriminated against accessing services, affecting their well-being and QoL [5]. Autonomy, the level of control that one has over its own life, and opportunities for social engagement and participation are vital for an individual’s health and well-being [9,10]. The effects of social inclusion on health and well-being are known to include behavioral and biological impacts and influence the level of poverty and education. Minorities and marginalized groups have the greatest hindrance to high levels of social inclusion [11].

In assessing SDG3 achievements, Sweden, Denmark, and the Netherlands ranked as the best countries in the EU between 2010 and 2019. Several indicators were assessed, including healthy life years (HLY), the share of people with good or very good perceived health, smoking prevalence, standardized death rate (for tuberculosis, HIV, and hepatitis), standardized preventable and treatable mortality, self-reported unmet need for medical examination and care, obesity rate, people killed in accidents at work, road traffic deaths, noise- and air pollution [12]. Despite this, there are questions if Sweden is the model welfare state it once was. In 2008 Sweden was the country in Europe with the lowest risk of poverty and social exclusion [2]. Ten years later, the risk of poverty for persons with disabilities has grown at one of the fastest rates in the EU. Higher cost of living and less financial support has been evident since 2008. Fifty percent of women with disabilities aged sixteen to twenty-five report difficulties making a living, compared with twenty-five percent of women without disabilities. The costs of rent, transportation, rehabilitation, dental care, medicine, and other expenses often exceed their income. There are also significant variances based on sex and geographical differences in the level of support provided by municipalities [2].

Quality of Life is a marker for understanding a population’s general health and satisfaction. The World Health Organization (WHO) defined health in 1948 as ‘a state of complete physical, mental and social well-being, not merely the absence of disease and impairments’ [13]. With this definition, measurements addressing health potential, well-being, and Quality of Life are needed. The overall focus tends to shift even more to health promotion and the well-being of populations rather than only reducing symptoms and illnesses [14]. It is crucial to understand more than why people die; we also need to know how people live with their health conditions [3]. The WHO definition of QoL used in this study includes both the subjectivity and context of the individual. The WHO defines QoL as ‘an individual’s perception of their position in life in the context of the culture and value systems in which they live and in relation to their goals, expectations, standards, and concerns’. This definition reflects that QoL refers to an individual appraisal rooted in a cultural, social, and environmental context. QoL cannot be explained only with ‘health status’, ‘lifestyle’, ‘life satisfaction, ‘mental state’, or ‘well-being’ [15].

Little is known about the QoL and how QoL is influenced by the social and economic situation of people with mobility-related disabilities in Sweden. Only a few studies on mobility-related disability and QoL in the Swedish context have been published within the last ten years [16,17,18,19,20,21,22,23]. International data have found a relationship between the perceived QoL and the level of disability [24,25,26,27,28,29,30,31,32], interaction and social activity [24,25,33,34], and increased age [24,30,35,36]. There also seems to be a relationship between QoL, education, and economic situation [25,28,33,34].

Therefore, the study’s objective was to explore if there were differences in QoL between a group with and without mobility-related disabilities in Sweden. Primary research questions were to assess if there were differences in General Health, General QoL, Health Satisfaction, and the four QoL domains of the WHOQOL-BREF (Physical Health, Psychological, Social Relationships, and Environment) between the groups. Secondary research questions were to explore if sex, age, level of education, economic situation, social inclusion, and the presence of additional or other than mobility-related disability predicted score change of the QoL items and domains of the WHOQOL-BREF.

## 2. Materials and Methods

### 2.1. Study Design, Participants, and Data Collection

The study was a comparative cross-sectional study where data was collected in an electronic self-reported questionnaire. 

Recruitment followed a convenient sampling strategy, and data were collected from 14 February to 31 March 2022. Recruitment advertisement was announced and promoted on various Facebook and social media platforms and was repeated multiple times to increase the number of responses. The advertisement for participants had a link to an electronic platform, QuestionPro (© 2022 QuestionPro Survey Software, APIv2). It was possible to read detailed information about the study and find contact information to project management on the start page. 

Inclusion criteria for the study were: Age over 18 years, willingness and ability to consent to participation (filling in a self-reported questionnaire), and ability to fill in surveys in Swedish. No additional inclusion nor exclusion criteria were applied.

This project was conducted following the ethical principles that have their origin in the Declaration of Helsinki [37] and further elaborated in the Belmont report [38] regarding respect for persons and their autonomy, beneficence, and justice. Furthermore, this study adheres to the STROBE statement, and the STROBE checklist was used for structuring the article [39]. Before data collection, approval from the Swedish Ethical Review Agency was obtained (Dnr 2021-06256-01). 

### 2.2. Instrument/Questionnaires 

The questionnaire used for data collection consisted of three main parts, demography and background, Quality of Life assessment, and a section with questions assessing social inclusion and economic situation. 

The demographic information collected was sex, age, geographical area of residence and living arrangements, foreign heritage, civil status, level of education, employment status, and experience of mobility-related disabilities or other disabilities or impairments. The question relating to the presence of mobility-related disability divided the participants into two main groups for the main comparisons. I.e., having a self-reported mobility-related disability (MOBDIS) or not (No-MOBDIS). 

The World Health Organization Quality of Life assessment, abbreviated version (WHOQOL-BREF), addressed the QoL. It is a 26-item version of the WHOQOL-100. The WHOQOL was developed by WHO and had a global definition of QoL and measured the perceived effects of disease and health interventions on the individual’s QoL. The WHOQOL and WHOQOL-BREF have been field-tested and validated in multiple countries and contexts. The WHOQOL-BREF produces four domain scores: Physical Health, Psychological, Social relationship, and Environment. Additionally, three items are examined separately: an individual’s overall perception of QoL (General QoL), their overall perception of their health satisfaction (HS), and a question about the current general health state (GH). Domain and item scores are scaled positively, i.e., each item is answered on a scale from 1 to 5, where a higher score denotes higher levels. The mean score of items within each domain was used to calculate the domain score [15]. The domain scores are presented as a score value between 4 and 20, and the item ratings are presented as a value between 1 and 5. Permission was obtained from the WHO to use the WHOQOL-BREF.

A Social Inclusion Score (SIS) was calculated to measure community inclusion and participation. SIS was based on seven questions deriving from community-based rehabilitation (CBR) indicators, covering interpersonal relationships and community participation, two domains important for understanding social inclusion [11]. The CBR indicators were developed by WHO and the International Disability and Development Consortium (IDDC) to examine differences in health, education, social life, and empowerment between individuals with and without disabilities within the same community [40]. The SIS is positively scaled and ranges between 7 and 35. This study used the SIS as a total score and a dichotomized variable. The CBR Indicators are not currently translated into Swedish but were translated into Swedish by the authors of this article.

The economic situation was assessed by an Economic Situation Score (ESS). This score was created for the study and consisted of five questions concerning lively hood and economic standing. The ESS is positively scaled and ranges between 5 and 25. Two of the questions in ESS originated from CBR indicators [40]. One of the questions, relating to the existence of a cash margin (12,000 SEK), originated from Statistics Sweden (Swedish: SCB), and one question was a demographic question used in other studies [41]. The last question was created for this study. The wording and scales for the ESS and SIS scores are available in Appendix A. 

### 2.3. Data Analysis

In line with the manual of WHOQOL-BREF, statistics were calculated as means and standard deviations of domains and item scores [15]. 

Datasets were included for analysis if they contained responses that fulfilled the requirements for calculating the WHOQOL-BREF domain scores, i.e., not more than one value was missing in each domain.

A two-sided Welch t-test, an unequal variances t-test [42,43], was used to compare the groups MOBDIS vs. No-MOBDIS, and the main groups stratified by sex, SIS, cash margin, education, and reports of additional or other disabilities for within and between-group differences. 

Dependent variables were dichotomized to examine differences. Social inclusion (SIS) was dichotomized to ‘lower’ and ‘higher’ using the median as the value cut. The level of education was dichotomized to ≤12 years of school or >12 years. The individual’s cash margin availability (Yes or No) was used to dichotomize the study participants for comparisons of QoL based on their economic standing. 

As the variable responses were both continuous and binary, generalized linear models (GLM) were applied to predict score change for the WHOQOL-BREF items and domains [44,45,46]. One model was used for the whole group to predict score change of the dependent variable WHOQOL-BREF items and domains with the independent variables for sex, age, education, group, SIS, cash margin, and additional or other disabilities. The selection of the independent variables for the model was predefined based on quantities of interest reported in previous studies on disability and QoL [5,6,24,25,26,27,28,30,33,34]. 

Data analyses were made with the computer program ‘R’ version 4.1.3 [47]. 

A *p*-value less than 0.05 was considered significant. No adjustments were made to the level of significance for the multiple comparisons due to the exploratory design of the study [48,49].

## 3. Results

### 3.1. Characteristics of the Study Group

The survey was initiated 470 times. Of those, 381 surveys were included for analyses. Eighty-nine were excluded from the analysis. Eighty-five did not have complete datasets for calculating WHOQOL-BREF scores. The incomplete surveys were a combination of on-lookers, i.e., consented but never started, and incomplete surveys. Four surveys were excluded from further analysis. One subject failed the inclusion criteria for age. Three subjects completed two survey rounds, and their last entry was excluded. 

One hundred forty-three participants reported mobility-related disabilities and were assigned to the group MOBDIS. The most reported reasons for mobility-related disability were chronic pain, neurological injury or disease, and muscular weaknesses. Two hundred thirty-eight participants reported no mobility-related disabilities, and they were assigned to the group No-MOBDIS. 

The age range for all participants was 20 to 94 years, with a total mean of 52 years. There were 317 female and 61 male participants. Three reported sex as ‘other’. Most of the participants had Swedish heritage. Both groups reported the civil status as ‘married/partner’ most frequently, but the No-MOBDIS group was proportionally larger. A higher proportion of the MOBDIS group reported living alone and reported a lower level of completed education.

In the MOBDIS group, 66.4% reported the presence of additional permanent disabilities, with severe pain being the most frequent, followed by chronic diseases and reduced vision. In the No-MOBDIS group, 39.7% reported experiencing disabilities other than mobility-related disabilities. For the No-MOBDIS group, mental ill-health was most frequent, followed by reduced vision and chronic disease. 

There were statistically significant differences (Chi-square test, *p* < 0.05) between the groups for sex, age, presence of other or additional disabilities, civil status, living situation, education, and cash margin. Additionally, there were statically significant differences at *p* < 0.001 level (Welch t-test (two-sided)) between the groups for SIS and ESS, where MOBDIS scored worse on all items/scores.

Characteristics of the study groups are presented in Table 1.

All 21 regions in Sweden were represented by respondents from both groups. Most respondents from the MOBDIS group came from the regions of Stockholm (24.6%), Västragötaland (12.7%), Skåne (8.3%), and Östergötland (7%). Most participants in the No-MOBDIS group came from Jämtland/Härjedalen (18.8%), Västernorrland (16.5%), Västragötaland (14.3%), and Stockholm (13.9%).

### 3.2. Quality of Life, between-Group Comparison

Scores were calculated for General Health (GH), General QoL, Health Satisfaction (HS), and the four domains of the WHOQOL-BREF (Physical Health, Psychological, Social Relationships, and Environment). Additionally, scores were calculated for SIS and ESS. These scores were compared between the group of MOBDIS and No-MOBDIS. The same set of scores was also compared for the group subset for sex (female or male), dichotomized SIS, cash margin (yes or no), educational level (≤12 years or >12 years), and presence of other or additional disability (yes or no).

There were significant differences between the groups found in all items; at a significance level of *p* < 0.001, apart from the Social relationship domain, which was significant at *p* = 0.002 (Table 2 presents WHOQOL-BREF items and Table 3 presents WHOQOL-BREF domains). The MOBDIS group scored lower on all seven items and domains. A lower score means a worse rating.

Subsetting the MOBDIS and No-MOBDIS groups to further explore group differences revealed similar patterns for the whole group, i.e., the MOBDIS group scored lower than the No-MOBDIS group. The between-group differences remained for GH, General QoL, HS, and the Physical domain, for group comparison subset to sex (female or male), dichotomized SIS (lower or higher), Cash margin (no or yes), Education (≤12 years or >12 years), and other/additional disability (yes or no). For the Psychological domain, the differences found remained for sex, education, and the existence of cash margin. In the Social domain, differences were found for sex, lower SIS, the existence of cash margin, and lower education. There were differences for all in the Environmental domain but not for the existence or not of other or additional disabilities (Table 2 and Table 3). 

The highest General QoL score (4.26 ± 0.64) was found for No-MOBDIS in the dichotomized SIS higher group. The lowest General QoL score (2.59 ± 1.05) was found in MOBDIS with no cash margin (Table 2).

### 3.3. Quality of Life, within-Group Associations

For within-group associations of both groups, there were statistically significant associations for dichotomized SIS, cash margin, and other/additional disability. The associations reflect those individuals with higher SIS, availability of cash margin, and no other/additional disability reported better mean scores for all items, indicating better perceived QoL (Table 4 presents WHOQOL-BREF items and Table 5 presents WHOQOL-BREF domains).

### 3.4. Variables Predicting WHOQOL-BREF Items and Domain Scores

Generalized linear models were created for WHOQOL-BREF items (GH, General QoL, HS) and the four domains of the WHOQOL-BREF (Physical Health, Psychological, Social Relationships, and Environment). The results of the model are presented in Table 6. The model contained the independent variables of sex, age, dichotomized education, group, SIS, Cash margin, and the existence of additional or other disability. The variables selected for the model were based on previous literature findings.

Looking at the whole dataset (*n* = 381), an increased SIS indicated higher scores for all items and domains (*p* ≤ 0.001). Group MOBDIS indicated a lower score for physical health and environmental domain and all items’ scores (*p* ≤ 0.001). Similar patterns were also seen for the additional/other disability question. Having an additional or other disability than mobility-related disability also had a negative predictive effect (*p* ≤ 0.001) on all items and all domain scores except the Social domain. The absence of a cash margin also seemed to negatively influence the GH, HS, Psychological domain, and Environment domain scores.

Sex and education level did not predict score change for the WHOQOL-BREF domains or items. Age only indicated a small negative effect for the item GH.

## 4. Discussion

This study performed in Sweden indicates that the group with mobility-related disabilities generally scored lower on the WHOQOL-BREF, i.e., having lower QoL than those without mobility-related disabilities. This trend remained at large when controlling for sex, dichotomized SIS, Cash margin, Education, and other/additional disability, for GH, General QoL, HS, and Physical domain.

Looking at the generalized linear model, the variables that seem to be most predictive of a negative score change were group MOBDIS, if additional or other disabilities were present, and the absence of a cash margin. 

This study indicates that having an additional disability to the mobility-related disability or having another disability than the mobility-related disability (for the No-MOBDIS group), the scoring was even lower on the different QoL ratings. This finding is in line with previous literature from international studies reporting on lower QoL scores for groups with more or a more severe level of disabilities [24,25,26,27,28,29,31,32]. As this present study had a cross-sectional design, changes in the prevalence of disabilities with age could not be assessed. Still, the group with mobility-related disabilities had a higher mean age and a higher proportion of additional/other disabilities than the No-MOBDIS groups. Even so, age did not significantly affect observed QoL, which was unexpected as a physical decline related to the aging process is expected. In addition, with increased age, an impairment can progress to a more severe stage or contribute to secondary negative health impacts, leading to higher levels of disabilities as perceived by the individual [3]. International research concerning mobility-related disability and QoL has seen patterns of an increased prevalence of disability as age increases and a higher level of disability for those with more chronic impairments or diseases [24,30,35]. 

Other studies have seen a relationship between QoL and education. Disability was more prevalent in those with lower education [25,28], and those with more severe disabilities were less likely to be employed [29]. The effect of education did not seem to largely influence the perceived QoL level in this study, even if the MOBDIS group had a generally lower level of education.

In this study, the economic situation or absence of a cash margin revealed lower QoL scores for both groups in the within-group comparison. This is consistent with previous studies where lower economic status seems to influence the perceived QoL [25,28,33,34]. Furthermore, in this study, the MOBDIS group had a significantly lower ESS score (18.45 vs. 21.09) and a higher proportion of persons who did not have a cash margin than the No-MOBDIS group (43.1% vs. 25.6%). These numbers are in line with numbers from larger censuses in Sweden. Statistics from 2019 reported a lack of cash margin in the general population, 16 years or older, of 17.1 (±0.4)%. The lack of cash margins for people with mobility-related disabilities was 37.9 (±4.2)% [50]. The higher proportion of people without cash margin in the mobility disability-related group is an important finding as people with disabilities experience financial barriers to health care even in high-income countries. One of the questions in the ESS score in this study asked if the participant had refrained from visiting a doctor or picking up medicine due to lack of money, and more people in the MOBDIS group reported that fact (Table 1). Despite Sweden being considered a highly developed Welfare state, where all citizens are supposed to have equal access to healthcare, as well as having generous general health insurance, the relatively low out-of-pocket cost (approximately 10–40 Euro per visit up to 120 Euro/year, then free [51]) appears to restrict the study participants with MOBDIS more than the No-MOBDIS from healthcare. The most significant financial barriers reported by WHO are out-of-pocket expenses, which are increasing and particularly impacting people with chronic diseases and disabilities [7]. The European Human rights reports suggest that countries should ensure that support services for the general population should be made accessible for persons with disabilities. Ensuring access to these mainstream services will reduce the need to turn to specialized services and minimize disability-related costs [2].

Other studies have revealed mixed results for sex and Quality of Life. When comparing the QoL of females with disabilities to those without, primarily the physical QoL domain seems to be affected [27,28]. A similar comparison reported significantly lower QoL scores for men without disabilities in the Physical health, Psychological, and Environmental domains [28]. The results were mixed when assessing QoL differences between sexes in the population with disabilities. Comparable QoL of the elderly was found in Poland [25]. Sex as a single factor did not explain differences in QoL among Finnish youths with severe physical disabilities [27]. For the stroke patients in Mongolia and the lower limb amputees in the USA and Tanzania, female patients had higher overall QoL than male patients [24,33]. In contrast, in Mexico, a population with neuromusculoskeletal or movement-related function disabilities, men reported a more favorable QoL even if the differences between sexes were minor [35]. This present study did not detect within-group differences based on sex. However, this result might also be influenced by the disproportion of sex representation in the study, as most participants were female. Another confounding factor may be asking for sex and not gender. QoL-related items might have been better assessed by the identity rather than male or female biological aspects.

Other factors that seem to influence the perceived QoL were interaction and social engagement, where those with more social interaction and engagement reported better QoL [24,25,33,34]. This pattern is also detected in this study as both groups revealed within-group differences for all domains and items, where the groups that scored higher on SIS also scored higher on the QoL ratings. For the between-group comparison, the SIS score had less impact on the social and psychological domains.

Cross-sectional data has design limitations and only reflects a point in time, as the exposure and outcome are simultaneously evaluated. Additionally, findings reflect correlational and not causal links [52]. In this case, the assessment of whether a person has a mobility-related disability, the economic situation, their perceived QoL, and social inclusion at the same time. No temporal relationship between these items can be explored, and it is impossible to determine the root cause of the outcome. A bidirectional relationship between poverty and disability is known, which likely affected the reported QoL outcome in this study. Understanding QoL is essential for improving care and has been used to identify the range of problems affecting patients. QoL has also been demonstrated as a strong predictor of survival [53]. 

In this study, the WHOQOL-BREF was the primary assessment for QoL. The WHOQOL-BREF is a commonly used tool for QoL assessment and is often used interchangeably with the Medical Outcome Study SF-36 (SF-36). Both instruments are used in relation to mobility-related disabilities, such as lower limb amputation, spinal cord injuries, and cervical spondylotic myelopathy (CSM) [54,55,56]. However, SF-36 and WHOQOL-BREF appear to measure distinct concepts related to QoL. It is suggested that WHOQOL-BREF measures the satisfaction of performing an activity, whereas the SF-36 measures the ability to perform the activity. The SF-36 seems to be more related to HRQOL [57]. The WHOQOL-BREF has a global definition of QoL, which matches the objectives of the study. The WHOQOL and WHOQOL-BREF have been reported to have good discriminant and content validity, internal consistency, and test-retest reliability in cross-sectional studies of adults in several different countries [14]. Cronbach’s alpha for this study was 0.81 for the Physical Health domain, 0.91 for the Psychological domain, 0.76 for the Social relationship domain, and 0.91 for the Environment domain. The relatively high Cronbach’s alpha indicates good internal consistency. The lower value for the social domain could be explained by the domain consisting of three questions. Calculations of Cronbach’s alpha are influenced by the number of items within each scale [58]. A lower Cronbach alpha for the Social relationship domain has been reported in other studies [15,28]. 

Social media platforms are becoming more used in health promotion activities and recruitment for studies. Research in the US has shown little difference in the proportions of US adults who use these platforms based on education level, income, or developed environment (i.e., urban vs. rural), indicating that this type of recruitment could have a broad reach. Despite its advantages, there are some limitations to social media recruitment, including the need for internet access and the overrepresentation of women. Social media platform recruitment might introduce sampling bias [53,58]. This present study was based on a convenience sample from recruitment through social media, which might affect the generalizability of the findings. In addition, most of the participants in this study were female, similar to other studies using social media recruitment [59,60]. 

With the open recruitment, the end range of the call’s reach for participants is unknown, as the recruitment call on social media platforms was shared multiple times. Therefore, it is difficult to say anything about the general survey response rate. Almost 18% of the initiated surveys were excluded from the data analysis due to incomplete data sets for WHOQOL-BREF score calculations. The non-complete data sets might be related to unwillingness to complete once started. Open recruitment also attracts “on-lookers”, i.e., persons who might be curious about the survey but have no intention of completing or even taking the survey. 

This study used a self-report questionnaire, which could lead to response bias. As this was an online survey, there was no control for if a respondent indeed had a mobility-related disability or not. It is unlikely that people misrepresent themselves in the survey, as there is no gain for the individual to score higher or lower than their perception of reality.

The scores used for assessing social inclusion, SIS, and economic standing, ESS, originated from CBR indicators. There are no validated Swedish versions of the CBR Indicators to the current date. The authors of this article did a translation. The validation of the translation was not within the scope of this project. As the origin of the CBR Indicators came from WHO, the general terms and concepts were similarly phrased to other WHO questionnaires validated for Swedish. Those questionnaires were used as guidance for terminology when translating the CBR indicators. 

The ESS, used to assess the economic standing, is not a validated instrument but rather questions from different contexts. The questions cover different areas of economic terms that are part of other instruments like the CBR indicators. The utilization of the cash margin as a variable for the economic standing is used in other censuses, for example, Statistics Sweden.

The decision to use 0.05 as the level of significance and not adjust that level for the multiple comparisons might introduce the risk of multiplicity. Results must be interpreted with this in mind. Still, the overall risk of multiplicity in these study results is considered low as most of the differences detected were at a significance level of 0.001 or lower. Furthermore, as this study was exploratory, there is no recommendation that adjustments for multiple comparisons are needed [48].

## 5. Conclusions

Despite limitations in the cross-sectional design to identify causal links as well as a risk for sampling bias, the result provides valuable explorative information that may guide future research on the topic. This study suggests an association between disability and QoL, showing that people with mobility-related disabilities have lower QoL than those without mobility-related disabilities. QoL was also related to lack of cash margin, a lower social inclusion score, and if there were additional or other disabilities present. Poverty and disability have a known bidirectional relationship. In. addition, this study performed in the welfare state of Sweden, the presence of a disability and a lower economic standing had a negative impact on QoL. Therefore, it is crucial to consider both aspects and facilitate social inclusion for better health equity. In any health promotion and community development-related work, and any related future studies, it is essential to understand and account for the causality, complexity, and relations of economy, social inclusion, and health conditions, even in a high-income country, to improve the health equity and Quality of Life among groups. 

## Figures and Tables

**Table 1 ijerph-19-15109-t001:** Characteristics of the study group.

		Mobility Disability *n* = 143	No Mobility Disability *n* = 238	Sign.
Sex (%)				0.016 ^a^
	Female	109 (76.2)	208 (87.4)	
	Male	32 (22.4)	29 (12.2)	
	Other	2 (1.4)	1 (0.4)	
Age, years [mean (SD)]		55.34 (13.44)	51.23 (14.20)	0.006 ^a^
Type of mobility-related disability (multiple-choice, count)				
	Amputation	15	-	
	Paralysis	15	-	
	Muscular weakness	33	-	
	Spinal cord injury	21	-	
	Neurologic injury or disorder	55	-	
	Cerebral Paresis	5	-	
	Chronic pain	72	-	
	Other reasons	30	-	
Use of assistive device (multiple-choice, count)				
	Wheelchair	50	-	
	Electric Wheelchair/Scooter (Permobil)	27	-	
	Prosthesis	12	-	
	Orthoses	32	-	
	Walker, walking table	15	-	
	Stick, crutch, cane	29	-	
	Other assistive devices	16	-	
Other/additional disability, yes (%)		95 (66.4)	94 (39.7)	<0.001 ^a,^**
Type of other/additional disability (multiple-choice, count)				
	Reduced vision	28	26	
	Blindness	1	0	
	Reduced hearing	15	15	
	Deafness	2	0	
	Severe pain	56	4	
	Mental ill-health	22	49	
	Cognitive disability	20	20	
	Chronic disease	40	22	
Foreign heritage (%)				0.070 ^a^
	One or both parents were born in Sweden	125 (87.4)	203 (85.3)	
	Both parents born outside Sweden	14 (9.8)	34 (14.3)	
	I do not know/Do not want to disclose	4 (2.8)	1 (0.4)	
Civil status (%)				0.003 ^a,^*
	Married/partner	68 (48.2)	152 (64.1)	
	Single	63 (44.7)	79 (33.3)	
	Other	10 (7.1)	6 (2.5)	
Living situation (%)				0.001 ^a,^*
	Living alone	59 (41.3)	74 (31.0)	
	Living with partner	53 (37.1)	103 (43.1)	
	Living with close family	22 (15.4)	61 (25.5)	
	Living with friends	1 (0.7)	0 (0.0)	
	Other	8 (5.6)	1 (0.4)	
Highest level of completed education (%)				<0.001 ^a,^**
	≤9 years	7 (4.9)	7 (2.9)	
	9–12 years	46 (32.2)	33 (13.8)	
	>12 years (university)	80 (55.9)	175 (73.5)	
	>12 years (other)	7 (4.9)	19 (8.0)	
	Other education	3 (2.1)	4 (1.7)	
Q: Have your health provider ever discussed the health benefits of eating healthy food, regular exercise, and not smoking? [Yes (%)]				0.744 ^a^
		88 (62.4)	150 (64.7)	
Q: Would you/your household be able to pay an unexpected expense of 12,000 SEK within the month without lending money or asking for help? [No (%)]				0.001 ^a,^*
		59 (43.1)	57 (25.6)	
Q: Has it happened during the last year that you have to refrain from visiting a doctor, dentist, picking up medicines, or aids because there was not enough money? [count, (%)]				0.005 ^a^
	Never	84 (61.3)	170 (76.2)	
	Occasionally	31 (22.6)	29 (13.0)	
	Often	8 (5.8)	17 (7.6)	
	Monthly	6 (4.4)	3 (1.3)	
	Several times per month	8 (5.8)	4 (1.8)	
Social inclusion score (SIS), [Total score (SD)]		26.57 (5.15)	29.50 (4.44)	<0.001 ^b,^**
Economic status score (ESS), [Total score (SD)]		18.45 (5.67)	21.09 (4.58)	<0.001 ^b,^**

^a^ Chi-square test, ^b^ Welch t-test (two-sided), * *p* < 0.05, ** *p* < 0.001.

**Table 2 ijerph-19-15109-t002:** Between-group comparison for WHOQOL-BREF items, [mean (SD)] *p*-value for Welch t-test (two-sided).

	General Health			General QOL			Health Satisfaction		
	MOBDIS	No MOBDIS	Sign.	MOBDIS	No MOBDIS	Sign.	MOBDIS	No MOBDIS	Sign.
Whole Group	2.87 (1.05)	3.71 (0.87)	<0.001 **	3.13 (1.13)	3.93 (0.90)	<0.001 **	2.41 (1.04)	3.37 (1.07)	<0.001 **
Female	2.85 (1.03)	3.67 (0.88	<0.001 **	3.12 (1.09	3.91 (0.88)	<0.001 **	2.37 (1.03)	3.33 (1.05)	<0.001 **
Male	2.97 (1.12)	4.03 (0.68)	<0.001 **	3.22 (1.24)	4.17 (0.80)	0.001 *	2.59 (1.07)	3.69 (1.14)	<0.001 **
SIS lower	2.54 (0.99)	3.18 (0.90)	<0.001 **	2.64 (1.00)	3.25 (1.00)	<0.001 **	2.06 (0.95)	2.68 (0.97)	<0.001 **
SIS higher	3.44 (0.93)	3.99 (0.73)	<0.001 **	3.92 (0.88)	4.26 (0.64)	0.015 *	2.96 (0.99)	3.74 (0.92)	<0.001 **
Cash margin no	2.53 (1.07)	3.35 (0.97)	<0.001 **	2.59 (1.05)	3.33 (0.99)	<0.001 **	2.10 (1.01)	2.95 (1.12)	<0.001 **
Cash margin yes	3.13 (0.97)	3.85 (0.79)	<0.001 **	3.51 (1.05)	4.13 (0.77)	<0.001 **	2.62 (1.05)	3.54 (0.99)	<0.001 **
Edu ≤ 12 year	2.74 (1.13)	3.70 (0.88)	<0.001 **	3.00 (1.16)	3.98 (1.10)	<0.001 **	2.42 (1.13)	3.50 (1.01)	<0.001 **
Edu > 12 year	2.94 (1.00)	3.71 (0.87)	<0.001 **	3.20 (1.12)	3.92 (0.85)	<0.001 **	2.39 (1.00)	3.34 (1.08)	<0.001 **
Other/additional disability	2.63 (1.02)	3.31 (0.90)	<0.001 **	2.86 (1.11)	3.52 (0.97)	<0.001 **	2.20 (0.95)	2.81 (1.04)	<0.001 **
No additional/no disability	3.33 (0.95)	3.97 (0.74)	<0.001 **	3.65 (1.00)	4.20 (0.74)	<0.001 **	2.81 (1.10)	3.73 (0.92)	<0.001 **

* *p* < 0.05, ** *p* < 0.001.

**Table 3 ijerph-19-15109-t003:** Between-group comparison for WHOQOL-BREF domains, [mean (SD)] *p*-value for Welch t-test (two-sided).

	Physical			Psychological			Social			Environmental		
	MOBDIS	No MOBDIS	Sign.	MOBDIS	No MOBDIS	Sign.	MOBDIS	No MOBDIS	Sign.	MOBDIS	No MOBDIS	Sign.
Whole Group	10.99 (2.50)	13.78 (2.42)	<0.001 **	12.55 (3.46)	13.99 (3.00)	<0.001 **	12.60 (3.79)	13.75 (2.88)	0.002 *	12.78 (3.28)	15.21 (2.64)	<0.001 **
Female	11.08 (2.50)	13.68 (2.42)	<0.001 **	12.50 (3.47)	13.86 (2.92)	0.001 *	12.80 (3.82)	13.72 (2.86)	0.028 *	12.83 (3.05)	15.14 (2.57)	<0.001 **
Male	10.95 (2.42)	14.66 (2.27)	<0.001 **	12.92 (3.49	15.20 (3.16)	0.010 *	11.96 (3.64)	14.07 (3.08)	0.017 *	12.91 (3.88)	15.95 (2.77)	0.001 *
SIS lower	9.86 (2.12)	11.91 (2.35)	<0.001 **	11.07 (3.10)	11.47 (3.02)	0.415	11.21 (3.49)	12.28 (2.69)	0.032 *	11.26 (2.77)	12.94 (2.68)	<0.001 **
SIS higher	12.73 (2.05)	14.63 (1.92)	<0.001 **	15.11 (2.47)	15.27 (2.10)	0.675	14.88 (3.09)	14.51 (2.68)	0.457	15.35 (2.22)	16.27 (1.90)	0.011 *
Cash margin no	9.84 (2.15)	12.83 (2.64)	<0.001 **	10.92 (3.37)	12.01 (3.20)	0.075	11.73 (3.74)	12.51 (3.11)	0.221	10.43 (2.84)	13.26 (2.76)	<0.001 **
Cash margin yes	11.77 (2.51)	14.06 (2.26)	<0.001 **	13.68 (3.18)	14.73 (2.59)	0.012 *	13.16 (3.79)	14.15 (2.73)	0.041 *	14.44 (2.52)	15.85 (2.30)	<0.001 **
Edu ≤ 12 year	10.44 (2.58)	14.06 (2.61)	<0.001 **	12.15 (3.32)	14.05 (3.23)	0.007	12.00 (3.83)	14.17 (2.86)	0.002	12.22 (3.09)	15.00 (3.01)	<0.001 **
Edu > 12 year	11.33 (2.43)	13.71 (2.39)	<0.001 **	12.79 (3.56)	13.97 (2.97)	0.007 *	12.99 (3.74)	13.66 (2.89)	0.136	13.10 (3.38	15.25 (2.57)	<0.001 **
Other/additional disability	10.41 (2.47)	12.72 (2.40)	<0.001 **	11.59 (3.23)	12.31 (2.90)	0.109	12.07 (3.82)	12.89 (2.93)	0.098	11.95 (3.27)	12.89 (2.93)	0.098
No additional/no disability	12.15 (2.15)	14.46 (2.19)	<0.001 **	14.43 (3.16)	15.10 (2.53)	0.185	13.64 (3.54)	14.32 (2.73)	0.226	14.41 (2.65)	14.32 (2.73)	0.226

* *p* < 0.05, ** *p* < 0.001.

**Table 4 ijerph-19-15109-t004:** Within-group associations for Sex, dichotomized SIS, cash margin, Education, other/additional disability WHOQOL-BREF items, [mean (SD)] *p*-value for Welch t-test (two-sided).

	General Health			General QOL			Health Satisfaction		
Female vs. Male	Female	Male	Sign.	Female	Male	Sign.	Female	Male	Sign.
Mobility disability	2.85 (1.03)	2.97 (1.12)	0.604	3.12 (1.09)	3.22 (1.24)	0.684	2.37 (1.03)	2.59 (1.07)	0.295
No Mobility disability	3.67 (0.88)	4.03 (0.68)	0.013 *	3.91 (0.89)	4.17 (0.80)	0.117	3.33 (1.05)	3.69 (1.14)	0.114
SIS Lower vs. higher	SIS lower	SIS higher	Sign.	SIS lower	SIS higher	Sign.	SIS lower	SIS higher	Sign.
Mobility disability	2.54 (0.99)	3.44 (0.93)	<0.001 **	2.64 (1.00)	3.92 (0.88)	<0.001 **	2.06 (0.95)	2.96 (0.99)	<0.001 **
No Mobility disability	3.18 (0.90)	3.99 (0.73)	<0.001 **	3.25 (1.00)	4.26 (0.64)	<0.001 **	2.68 (0.97)	3.74 (0.92)	<0.001 **
Cash margin yes vs. No	No	Yes	Sign.	No	Yes	Sign.	No	Yes	Sign.
Mobility disability	2.61 (1.01)	3.21 (1.02)	0.001 *	2.70 (1.06)	3.69 (0.99)	<0.001 **	2.11 (0.98)	2.78 (1.04	<0.001 **
No Mobility disability	2.53 (1.07)	3.13 (0.97)	<0.001 **	2.59 (1.05)	3.51 (1.05)	<0.001 **	2.10 (1.01)	2.62 (1.05)	0.004 *
Edu ≤ 12year vs. >12 year	Edu ≤12year	Edu >12 year	Sign.	Edu ≤12year	Edu >12 year	Sign.	Edu ≤12year	Edu >12 year	Sign.
Mobility disability	2.74 (1.13)	2.94 (1.00)	0.272	3.00 (1.16)	3.20 (1.12)	0.311	2.42 (1.13)	2.39 (1.00)	0.908
No Mobility disability	3.70 (0.88)	3.71 (0.87)	0.945	3.98 (1.10)	3.92 (0.85)	0.782	3.50 (1.01)	3.34 (1.08)	0.372
Additional or other disability	Yes	No	Sign.	Yes	No	Sign.	Yes	No	Sign.
No Mobility disability	2.63 (1.02)	3.33 (0.95)	<0.001 **	2.86 (1.11)	3.65 (1.00)	<0.001 **	2.20 (0.95)	2.81 (1.10	0.002 *
Mobility disability	3.31 (0.90)	3.97 (0.74)	<0.001 **	3.52 (0.97)	4.20 (0.74)	<0.001 **	2.81 (1.04)	3.73 (0.92)	<0.001 **

* *p* < 0.05, ** *p* < 0.001.

**Table 5 ijerph-19-15109-t005:** Within-group associations for Sex, dichotomized SIS, cash margin, Education, other/additional disability WHOQOL-BREF domains, [mean (SD)] *p*-value for Welch t-test (two-sided).

	Physical			Psychological			Social			Environmental		
Female vs. Male	Female	Male	Sign.	Female	Male	Sign.	Female	Male	Sign.	Female	Male	Sign.
Mobility disability	11.08 (2.50)	10.95 (2.42)	0.791	12.50 (3.47)	12.92 (3.49)	0.550	12.80 (3.82)	11.96 (3.64)	0.263	12.83 (3.05)	12.91 (3.88)	0.924
No Mobility disability	13.68 (2.42)	14.66 (2.27)	0.038	13.86 (2.92)	15.20 (3.16)	0.038	13.72 (2.86)	14.07 (3.08)	0.566	15.14 (2.57)	15.95 (2.76)	0.144
SIS Lower vs. higher	SIS lower	SIS higher	Sign.	SIS lower	SIS higher	Sign.	SIS lower	SIS higher	Sign.	SIS lower	SIS higher	Sign.
Mobility disability	9.86 (2.12)	12.73 (2.05)	<0.001 **	11.07 (3.10)	15.11 (2.47)	<0.001 **	11.21 (3.49)	14.88 (3.09)	<0.001 **	11.26 (2.77)	15.35 (2.23)	<0.001 **
No Mobility disability	11.91 (2.35)	14.63 (1.92)	<0.001 **	11.47 (3.02)	15.27 (2.10)	<0.001 **	12.28 (2.69)	14.51 (2.68)	<0.001 **	12.94 (2.68)	16.27 (1.90)	<0.001 **
Cash margin yes vs. No	No	Yes	Sign.	No	Yes	Sign.	No	Yes	Sign.	No	Yes	Sign.
Mobility disability	10.08 (2.33)	12.15 (2.23)	<0.001 **	11.21 (3.39)	14.26 (2.87)	0.001 **	11.67 (3.87)	13.75 (3.40)	0.001 **	11.02 (2.89)	15.04 (2.27)	0.001 **
No Mobility disability	9.84 (2.15)	11.77 (2.51)	<0.001 **	10.92 (3.37)	13.68 (3.18)	0.001 **	11.73 (3.74)	13.16 (3.79)	0.029 *	10.43 (2.84)	14.444 (2.52)	0.001 **
Education ≤12year vs. >12 year	Edu ≤12 year	Edu >12 year	Sign.	Edu ≤ 12 year	Edu >12 year	Sign.	Edu ≤ 12 year	Edu > 12 year	Sign.	Edu ≤ 12 year	Edu > 12 year	Sign.
Mobility disability	10.44 (2.58)	11.33 (2.43)	0.045 *	12.15 (3.32)	12.79 (3.56)	0.286	12.00 (3.83)	12.99 (3.74)	0.136	12.22 (3.09)	13.10 (3.38)	0.114
No Mobility disability	14.06 (2.61)	13.71 (2.39	0.446	14.05 (3.23)	13.97 (2.97)	0.890	14.17 (2.86)	13.66 (2.89)	0.310	15.00 (3.01)	15.25 (2.57)	0.631
Additional or other disability	Yes	No	Sign.	Yes	No	Sign.	Yes	No	Sign.	Yes	No	Sign.
No Mobility disability	10.41 (2.47)	12.15 (2.15)	<0.001 **	11.59 (3.23)	14.43 (3.16)	<0.001 **	12.07 (3.82)	13.64 (3.54)	0.017 *	11.95 (3.27)	14.41 (2.65)	<0.001 **
Mobility disability	12.72 (2.40)	14.46 (2.19)	<0.001 **	12.31 (2.90)	15.10 (2.53)	<0.001 **	12.89 (2.93)	14.32 (2.73)	<0.001 **	13.95 (2.73)	16.02 (2.24)	<0.001 **

* *p* < 0.05, ** *p* < 0.001.

**Table 6 ijerph-19-15109-t006:** Prediction of score change between independent variables and WHOQOL-BREF item and domain scores, whole group, generalized linear model.

Independent Variables	WHOQOL-BREF Items			WHOQOL-BREF Domains			
	General Health	General QoL	Health Satisfaction	Physical Health	Psychological	Social Relationship	Environment
Intercept	1.982 **	1.252 **	0.600	5.981 **	3.833 **	5.634 **	5.867 **
Sex (M)	0.161	0.076	0.126	0.339	0.514	−0.102	0.165
Age	−0.007 *	−0.003	0.004	0.002	0.010	−0.014	0.004
Education (≥12 years)	−0.014	0.104	0.192	0.039	0.055	0.042	−0.081
Group (Mobility disability)	−0.451 **	−0.37 **	−0.648 **	−1.775 **	−0.008	0.012	−0.931 **
SIS	0.079 **	0.104 **	0.093 **	0.272 **	0.355 **	0.314 **	0.333 **
Cash margin (N)	−0.192 *	−0.408 **	−0.079	−0.279	−1.015 **	−0.351	−1.696 **

* *p* < 0.05, ** *p* < 0.001.

## Data Availability

Detailed reports concerning the underlying data of this study can be requested from the corresponding author.

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
