# Peer review of "Quality of Life of People with Mobility-Related Disabilities in Sweden: A Comparative Cross-Sectional Study"

_ijerph, 2022, doi:10.3390/ijerph192215109_

Round 1

Reviewer 1 Report

This study is enlightening. The kinds of data that the study provides will be helpful to both census collectors and academics working in the field of disability studies. Concrete data, as mined in this study, while subjective, bolsters our work to ensure that all members of our communities have the means to thrive.

Author Response

Thank you for your supportive words!

Language has been speel checked.

Reviewer 2 Report

Manuscript under review is a Quality of Life study aimed at a certain demographic located in Sweden. Authors collect the data via survey and study is especially important since data basis comes from a social welfare state. Authors provide their survey as a supplement file which is very valuable. Literature review covers a good portion of recent and relevant domain of the subject. I believe that paper has merit and suitable for publication in IJERPH if authors consider the following minor issues.

- Attributes of survey data is based on The World Health Organization Quality of Life assessment scheme. I understand that this is a validated and proven system. However, I still believe that alternatives to these attributes can be elaborated in a comparative manner (at least on an expected differences level)

- Authors provide an in-depth statistical analysis and a good discussion. Compared to these, conclusion section is very weak. I would definitely expect some thoughts on sampling issues (such as size or selection) and direction for future studies with different national characters.

- A very minor issue but illustration of p-value on tables is very annoying. Authors can delete usage of “p” on every line and just illustrate the score with relevant sign.

Author Response

Thank you for your valuable review!

Reviewer 2: Asked for:

  1. Minor spelling check
    1. Response: It has been done
  2. “Attributes of survey data is based on The World Health Organization Quality of Life assessment scheme. I understand that this is a validated and proven system. However, I still believe that alternatives to these attributes can be elaborated in a comparative manner (at least on an expected differences level)”
    1. Response: For general QoL measurements (not disease specific tools) SF 36 and WHOQOL is often used for this demographic, but they measure different aspects of QoL. A comparison to the SF 36 is added to the discussion. And new references are added in relation to this section.
  3. “Authors provide an in-depth statistical analysis and a good discussion. Compared to these, conclusion section is very weak. I would definitely expect some thoughts on sampling issues (such as size or selection) and direction for future studies with different national characters.”
    1. Response: We have tried to make the conclusion more sharp with regards to the limitations of the cross sectional data. Regarding Sampling issues: In lines 425-442, information regarding challenges and possible sampling bias is discussed.
  4. “A very minor issue but illustration of p-value on tables is very annoying. Authors can delete usage of “p” on every line and just illustrate the score with relevant sign.”
    1. Response: Agree, we have adjusted it accordingly.

Reviewer 3 Report

Congratulations for the article. It is an interesting and attractive topic that provides information of interest to the scientific community.

It needs some revisions before being published.

-The title correctly defines the content of the article.

-The abstract is adequately structured.

-The keywords need to be revised. It is curious that the key word "mobility" does not appear when it is an important aspect of the work.

-The introduction and methodology are complete and extensive. They correctly describe the necessary information.

- The results are presented with tables. Section 3.3. Quality of Life, within-group comparison, needs a more detailed explanation.

-The discussion should be revised. In this section, the results obtained should be compared with the rest of the scientific literature published on the chosen topic. Sometimes, it seems that the information discussed could belong in the introduction section.

- The section 4.1. Methodological considerations should not be included in the discussion. Perhaps it would be more appropriate to include it in the methodology.

- I would add a section on future lines of research derived from this work.

-Revise the bibliography according to the bibliographic style of this journal.

-The supplementary material is correct.

Author Response

Thank you for you detailed review!

Reviewer 3: Comments and ask for the following to be developed/changed

  1. “The keywords need to be revised. It is curious that the key word "mobility" does not appear when it is an important aspect of the work.”
    1. Response: Keyword added as suggested. Usually, you try to avoid repeating words that are already included in the title, but we follow your recommendation
  2. “The results are presented with tables. Section 3.3. Quality of Life,within-group comparison, needs a more detailed explanation.”
    1. Response: Text has been revised to makes this clearer in lines 277-283 and in table 4 and 5
  3. “The discussion should be revised. In this section, the results obtained should be compared with the rest of the scientific literature published on the chosen topic. Sometimes, it seems that the information discussed could belong in the introduction section.”
    1. Response: We do not fully agree with this comment, as the results already systematically is compared with national and international literature on the topic. Mostly international due to lack of Swedish research/data.
  4. The section 1. Methodological considerationsshould not be included in the discussion. Perhaps it would be more appropriate to include it in the methodology.
    1. Response: The last 10-15 years it has become more and more common to perform cross-sectional survey by using online media platforms for recruitment, this poses large challenges since you have minor possibilities to have control over response rates, sample power and other forms of sampling bias. We find it important to bring these issues forward openly in the discussion for other researchers to learn from. Further, reviewer 2 has exactly asked for more discussion on sampling issues as already written in this section. We have revised text slightly, and removed the heading “Methodological considerations”
  5. “I would add a section on future lines of research derived from this work”
    1. Response: in the conclusion clearer statements are made about the limitations of cross sectional data, and the importance of understanding the causal relationship between the socioeconomic, disability and quality of life.
  6. “Revise the bibliography according to the bibliographic style of this journal.”
    1. Response: Style updated